# Analysis of the Influence of Ferromagnetic Material on the Output Characteristics of Halbach Array Energy-Harvesting Structure

**DOI:** 10.3390/mi12121541

**Published:** 2021-12-11

**Authors:** Xiangyong Zhang, Haipeng Liu, Yunli He, Tingrui Peng, Bin Su, Huiyuan Guan

**Affiliations:** 1School of Mechatronical Engineering, Beijing Institute of Technology, Beijing 100081, China; zhangxiangyong177@163.com (X.Z.); 3120200131@bit.edu.cn (T.P.); subinbaiyang@163.com (B.S.); ghy907553326@163.com (H.G.); 2State Key Laboratory of Explosion Science and Technology, Beijing Institute of Technology, Beijing 100081, China; 3Department of Mechanical Engineering, Shandong Huayu University of Technology, Dezhou 253034, China; heyunli1986@163.com

**Keywords:** Halbach array, iron sheet, energy harvesting

## Abstract

Due to the particular arrangement of permanent magnets, a Halbach array has an significant effect of magnetism and magnetic self-shielding. It can stretch the magnetic lines on one side of the magnetic field to obtain an ideal sinusoidal unilateral magnetic field. It has a wide application range in the field of energy harvesting. In practical applications, magnetic induction intensity of each point in magnetic field is not only related to the induced current and conductor but also related to the permeability of the medium (also known as a magnetic medium) in the magnetic field. Permeability is the physical quantity that represents the magnetism of the magnetic medium, which indicates the resistance of magnetic flux or the ability of magnetic lines to be connected in the magnetic field after coil flows through current in space or in the core space. When the permeability is much greater than one, it is a ferromagnetic material. Adding a ferromagnetic material in a magnetic field can increase the magnetic induction intensity B. Iron sheet is a good magnetic material, and it is easy to magnetize to generate an additional magnetic field to strengthen the original magnetic field, and it is easy to obtain at low cost. In this paper, in order to explore the influence of ferromagnetic material on the magnetic field and energy harvesting efficiency of the Halbach array energy harvesting structure, iron sheets are installed on the periphery of the Halbach array rotor. Iron sheet has excellent magnetic permeability. Through simulation, angle between iron sheet and Halbach array, radian size of iron sheet itself and distance between iron sheet and Halbach array can all have different effects on the magnetic field of the Halbach array. It shows that adding iron sheets as a magnetic medium could indeed change the magnetic field distribution of the Halbach array and increase energy harvesting efficiency. In this paper, a Halbach array can be used to provide electrical power for passive wireless low-power devices.

## 1. Introduction

With the continuous development of micro devices, various wearable devices that can be used on the human body have been widely used. Zhihao Ren et al. based one on the smart cut process for fabrication of silicon-on-insulator (SOI) wafers and films made of wafer bonding technology, which can satisfy high-performance molecular sensors for next-generation applications beyond 5G [1] At the same time as the development of micro devices, energy harvest technology has also been developed. Xiang Lu et al. designed an energy trap based on the principle of piezoelectricity and the miniature deformable squama mechanics [2] by controlling the size of the signal input to each array to produce different shapes of deformation to fit the human body movement. The piezoelectric energy harvest voltage is large, but it has strict requirements on frequency and bandwidth.

In 1979, physicist Klaus Halbach discovered a particular permanent magnet array—the Halbach array [3,4,5]. Using a special magnet unit arrangement, field strength in the magnet unit direction increased. The goal is to use a small number of magnets to generate the strongest magnetic field. The study by Ni Y, Xu L, Jing L et al. showed that the Halbach array has a higher air gap magnetic flux density amplitude than conventional magnetic steel, which can significantly improve the energy harvesting efficiency [6,7,8].

It is crucial to study the distribution of the magnetic field in electromagnetic harvesting energy, and it is the prerequisite for applying a magnetic medium in a magnetic field to improve the magnetic field [9,10,11,12]. After ferromagnetic material that is magnetized in the magnetic field, an additional magnetic field will be generated. The superposition of additional magnetic field and original magnetic field makes the total magnetic field much stronger than the original magnetic field [13,14,15]. In this paper, iron sheets are used to explore the influence of ferromagnetic material on the magnetic field distribution of the Halbach array. As the position, size and angle of iron sheet are different, the final magnetic field distribution of the Halbach array will be different, and the energy harvesting efficiency will be different. Three sets of simulation comparisons are designed for verification. The results prove that when it is a ring-shaped Halbach array composed of three pole pairs and two permanent magnets per pole, and the angle between the iron sheet and Halbach array is 30°, the magnetic field distribution is the densest and the energy harvesting effect is the best. When the radian of the iron sheet is 40°, the amount of iron sheet is the lowest, while the energy harvesting effect is the best. When the distance between the iron sheet and Halbach array gets closer, the energy-harvesting effectiveness of the designed structure grows higher. Compared with the case of without iron sheet, the maximum energy harvesting voltage is increased by 1.8 times, and the maximum energy harvesting voltage is 1.8 V. The simulation results are verified through experiments, and when iron sheets are added, the energy harvesting effect is indeed greatly improved.

## 2. The Principle of Halbach Array and Ferromagnetic Magnetization

In order to make the air gap flux density of the Halbach array magnet rotor present a sinusoidal distribution, it is necessary to perform continuous sinusoidal magnetization of the permanent magnets to make up the Halbach array rotor, namely Halbach magnetization [16]. The ideal sinusoidal magnetization effect is better, but the process is complicated to achieve with the existing technology. Therefore, the permanent magnets are assembled and magnetized in segments. According to the calculus theory, as long as the permanent magnets are magnetized and discretely connected as much as possible on each pole pair of the Halbach array, an approximately sinusoidal air gap magnetic density waveform can be obtained [17,18]. In this paper, the Halbach array composed of three pole pairs and two permanent magnets per pole needs to be magnetized and arranged according to the following equations.
(1)$$\beta =\frac{\left(p-1\right)\pi }{mp}$$
(2)$$\theta =\frac{\pi }{mp}\;$$
where m is the number of pieces of the permanent magnet under each pole, θ is the angle of a single permanent magnet, β is the magnetizing angle of two adjacent permanent magnets in a single pole and p is the number of motor pole pairs.

The permanent magnet arrangement of Halbach array used in this paper are shown in Figure 1.

For the Halbach array rotor with two permanent magnets per pole, each individual permanent magnet of the Halbach array is magnetized in a single direction, arranged in the radial and circumferential directions. The radially arranged permanent magnets mainly produce magnetic poles, and the tangentially arranged permanent magnets mainly guide and enhance the magnetic flux [19,20,21]. Because the unique structure of Halbach array can enhance the magnetic field strength on one side, its energy harvesting effect will also be better.

The Halbach array generates a sinusoidal magnetic field. When a ferromagnetic material is placed in the magnetic field, the ferromagnetic material is magnetized to produce an additional magnetic field. Magnetized body current $J_{m}$ appears in the ferromagnetic material after magnetization; Magnetizing surface current $K_{m}$ appears on the ferromagnetic surface.
(3)$$J_{m}=\nabla \times M.$$
(4)$$K_{m}=M\times e_{n}.$$
where M is the magnetization and $e_{n}$ is the unit vector of the outer normal of ferromagnetic surface.

Using Stokes’ theorem, from Equation (3) we can get
(5)$$\oint_{l}M\cdot dl=\int_{l}\left(\nabla \times M\right)\cdot dS=I_{m}$$
where $I_{m}$ is the magnetizing current.

The magnetic field in the presence of ferromagnetic material can be regarded as a synthetic magnetic field created by the original magnetic field and the magnetizing current. It can be obtained from the Ampere’s loop law.
(6)$$\oint_{l}B\cdot dl=\mu_{0}\sum I_{m}+B_{0}$$
where $B_{0}$ is the magnetic field strength of original magnetic field.

Therefore, it can be seen that adding ferromagnetic material in the magnetic field will indeed affect the original magnetic field. In this paper, the Halbach array produces a sinusoidal magnetic field, and the ferromagnetic material has different magnetization effects when placed in different positions.

Introduce a new field quantity—magnetic field strength H
(7)$$H=\frac{B}{\mu_{0}}-M$$
where $\mu_{0}$ is the vacuum permeability, and B is the synthetic magnetic field of original magnetic field and additional magnetic field generated by the ferromagnetic material.

For most ferromagnetic material, the magnetization is proportional to the magnetic field strength.
(8)$$M=\chi_{m}H$$
where $\chi_{m}$ is the magnetic susceptibility, which is a pure one-dimensional number. Substituting Equation (8) into (7), we can get
(9)$$B=\mu_{0}\left(1+\chi_{m}\right)H$$
(10)$$\mathrm{Let}\;\mu =\mu_{0}\left(1+\chi_{m}\right)$$

Get the combinatorial equation of ferromagnetic material
(11)B=μH
where μ is the permeability of ferromagnetic material.

It can be seen from the above that when ferromagnetic material is added to the magnetic field of Halbach array rotor, the physical properties of ferromagnetic material itself will also affect the magnetization.

## 3. The Effect of Iron Sheet on Halbach Array

In this paper, the influence of iron sheet on the Halbach array from angle between iron sheet and Halbach array, radian size of iron sheet and distance between iron sheet and Halbach array is explored. The Halbach array has three pairs of poles; the six pieces of iron sheet are evenly arranged on the outside of the Halbach array rotor. The other parameters are shown in Table 1.

The magnetic field distribution diagram of the Halbach array when there is no iron sheet is shown in Figure 2.

It can be seen from Figure 2 that the sinusoidal distribution of the magnetic field is better when there is no iron sheet. Using 6 groups of 100-turn coils, gap between the coil and the outer periphery of Halbach array rotor is 1 mm, and iron sheet is placed on the periphery of the coil with a gap of 0.5 mm from the coil. The radian of iron sheet is 40°for each piece, and the rotation rate is 10 r/s. The maximum energy harvesting voltage obtained by simulation is 1 V.

### 3.1. The Influence of the Angle between Halbach Array and Iron Sheet

In this paper, the coil is set to be fixed. The permanent magnet and the iron sheet are fixed as a whole to rotate together, and the positions of the permanent magnet and the iron sheet do not change during the rotation. The actual device is shown in Figure 3b

The Halbach array used in this paper is a three-pole pair. Thus, there are six rectangular permanent magnets arranged radially. There are six identical iron sheets and the distance between iron sheet is 3 mm from the Halbach array. In order to explore the relationship between the angle between iron sheet and the Halbach array, the angle between the iron sheet and the Halbach array is defined as the angle between the center of the iron sheet and the centerline of radially arranged permanent magnets with the magnetic poles, as shown in Figure 3a.

It can be seen from the Figure 3b,c that the relative position of the iron sheet and the Halbach array rotor is different, and it can be seen from the Figure that the angle of 60° between iron sheet and the Halbach array forms a circle. Therefore, the magnetic field distribution of the Halbach array under different conditions from 0 to 60° is explored, as shown in Figure 4.

It can be seen from Figure 4 above that the difference in the angle between iron sheet and the Halbach array will indeed cause the difference in the magnetic field distribution of the Halbach array. When the angle is 0°, the change in the magnetic field of the Halbach array by iron sheet is not apparent. Compared with no iron sheet, the magnetic lines are concentrated only in the gap between these two iron sheets. When the angle increases to 10°, the magnetic lines begin to stretch into iron sheet, and the magnetic field begins to gather in the gap between the iron sheet and the Halbach array. When the angle is 20°, the magnetization effect is more prominent, and the distribution of magnetic lines outside iron sheets is reduced. When the angle is 30°, magnetizing effect of iron sheet on the Halbach array reaches the maximum. The magnetic flux density in the gap is greatly enhanced, magnetic lines outside iron sheets are greatly reduced, and the magnetization effect is optimal. When the angle continues to increase, the magnetization effect begins to weaken. When the angle is 40°, the magnetic field distribution is the same as when it is 20°. When the angle is 50°, the magnetic field distribution is the same as when it is 10°. When the angle changes from 0 to 60°, its magnetization effect first increases and then decreases. To verify these conclusions, the energy harvesting simulation is carried out. The maximum voltage obtained under the same conditions as shown in Figure 5.

It can be seen from Figure 5 that when angle between the iron sheet and the Halbach array rotor is 0°, the energy harvesting effect is the worst. The peak-to-peak voltage is 1.2 V, which is an increase of 20% compared with case of the non-iron sheet. When angle increases, it is consistent with the magnetic field distribution result. Due to the concentration of iron sheet on magnetic field, the magnetic lines are stretched between the iron sheets and the Halbach array rotor, so its energy harvesting effect is greatly improved. When the angle is 30°, the peak-to-peak value of the harvesting energy voltage reaches the maximum of 1.8 V, which increases 80% compared to the case without iron sheets. When the angle exceeds 30°, the peak-to-peak voltage weakens. It is basically consistent with the change in magnetic field distribution. Between 0 and 60°, the peak-to-peak energy harvesting voltage and the magnetic field distribution are symmetrical at about the angle of 30°. It shows that when the angle between the iron sheet and the Halbach array rotor is 30°, the energy harvesting effect is the best.

### 3.2. The Influence of the Radian Size of the Iron Sheet and the Halbach Array

To explore the influence of the radian of the iron sheet on the Halbach array, the radian of the iron sheet is defined, as shown in Figure 6a.

It can be seen from the Figure 7b,c that the degree to which the iron sheet surrounds the Halbach array rotor is different, and it has been known from the above that when the angle between the iron sheet and the Halbach array is 30°, the energy harvesting effect is the best. Therefore, to explore the influence of the radian size of iron sheet on the Halbach array, the distance between the iron sheet and the Halbach array is 3 mm. The angle between the iron sheet and the Halbach array is 30°. The distribution of the Halbach array magnetic field under different radian of iron sheet is shown in Figure 7.

As you can see from Figure 7 above, when the angle between the Halbach array and the iron sheet is 30° and distance between the Halbach array and the iron sheet is 3 mm, the different sizes of iron sheet will result in different magnetic field distributions of the Halbach array. When the radian of each iron sheet is 60°, the iron sheet completely surrounds the periphery of the Halbach array rotor, the magnetic lines are compressed between the iron sheet and the Halbach array rotor, and the magnetic flux density modulus is the largest. When the radian of the iron sheet is 50°, it is the same as when the iron sheet is 60°. When the radian of iron sheet continues to decrease to 40°, the magnetic field distribution begins to change, iron sheet cannot completely surround the magnetic lines, and a small part of the magnetic lines stretch to the periphery of the iron sheet. When the radian of the iron sheet continues to decrease, the trend of change begins to increase, magnetic lines outside the iron sheet began to increase, and magnetic field density inside the iron sheet began to decrease. When the radian of the iron sheet is reduced to 15°, compared with the Halbach array magnetic field when there is no iron sheet, the magnetization effect only occurs in the iron sheet, and the magnetic flux density modulus is significantly reduced. In order to verify these conclusions, the energy harvesting simulation is carried out. Under the same conditions and under different iron radians, the maximum energy harvesting voltage obtained is shown in Figure 8.

It can be seen from Figure 8, when the radian of the iron sheet is between 60° and 40°, the energy harvesting effect does not change much, which is consistent with the magnetic field distribution results obtained above. The maximum energy harvesting voltage is 1.8 V. When radian of the iron sheet continues to decrease due to the weakening of the magnetic accumulation effect, its maximum energy harvesting voltage begins to decrease. The smaller the radian of the iron sheet, the lower its maximum energy harvesting voltage. When the radian of the iron sheet is reduced to 15°, its maximum energy harvesting voltage is 1.21 V. It shows that when the iron sheet and the Halbach array are at the same angle and distance, and radian of the iron sheet is 40°, the amount of iron sheet is the least and the energy harvesting effect is the best.

### 3.3. The Influence of the Distance between the Iron Sheet and the Halbach Array

The difference in the distance between the iron sheet and the Halbach array will lead to the difference in its magnetization effect. Distance between the iron sheet and the Halbach array is defined as the distance from the periphery of the Halbach array rotor to the iron sheet, as shown in Figure 9a.

It can be seen from the Figure 10b,c that only the distance between the iron sheet and the Halbach array is changed. When the angle between the fixed iron sheet and the Halbach array is 30°, and the radian of each iron sheet is 40°. If only changing the distance between the iron sheet and the Halbach array rotor, the magnetic field distribution diagram under different conditions is shown in Figure 10.

It can be seen from Figure 10 that when the distance is 3 mm, the magnetic lines between the iron sheet and the Halbach array are highly concentrated and densely distributed. As the distance increases, the magnetic lines still gather between the iron sheet and the Halbach array. Still, the concentration of magnetic lines gradually decreases, and the distribution becomes more and more sparse. Compared with the distance of 3 mm, when the space is 10 mm, there is a significant difference in the density of magnetic lines. It shows that the distance between the iron sheet and the Halbach array does cause a difference in the magnetic field distribution. In order to verify the influence of the distance between the iron sheet and the Halbach array on the energy harvesting effect, the energy harvesting simulation was carried out under the same condition, and the maximum energy harvesting voltage at a different distance is shown in Figure 11.

It can be seen from Figure 11 that as the distance between the iron sheet and Halbach array increases, the maximum energy harvesting voltage decreases. When the distance is 3 mm, the maximum energy harvesting voltage is 1.8 V, and the change in the distance and the maximum energy harvesting voltage is inversely proportional. When the distance just starts to increase, the maximum energy harvesting voltage drops extremely fast. When the distance continues to increase, the maximum energy harvesting voltage decreases slowly. When the distance is increased to 11 mm, the maximum energy harvesting voltage is basically maintained at about 1.07 V. Compared with the case without an iron sheet, the increase in energy harvesting effect is not obvious. It shows that in order to increase the energy harvesting effect, the distance between the iron sheet and the Halbach array should be reduced as much as possible.

## 4. Experimental Verification

From the above analysis, it can be known that the iron sheet has an important influence on the distribution of the magnetic field and energy harvesting effect of the Halbach array harvester. The angle between the iron sheet and the Halbach array, the radian of the iron sheet and the distance between the iron sheet and the Halbach array are all important for the energy harvesting effect. A set of comparative experiments was carried out. The coil used six groups of 100 turns each; the gap between the coil and the outside of the Halbach array rotor was 1 mm. The coil was located between the iron sheet and the Halbach array, and the distance between the iron sheet and Halbach array was 7 mm, and the rotation rate of Halbach array was 10 r/s. The experimental set up used in this paper is shown in Figure 12.

In order to verify the availability of energy harvesting of the Halbach array harvester, the Halbach array without iron sheets was selected to be tested through supplying energy to several low power devices, such as timers and thermometers. The arrangement of Halbach array and coils are shown in Figure 12 above. The gap between the coil and the outside of Halbach array rotor was 1 mm.

As shown in Figure 13, the Halbach array harvester can fully meet the power supply for various low power devices.

As can be seen from Figure 14, the measured voltage reached the maximum in extreme time without an iron sheet in the structure and when the angle between the fixed iron sheet and the Halbach array was 30° and the distance between the iron sheet and the Halbach array was 7 mm. The measured result verified the simulation results.

As can be seen from Figure 15, the relationship between the radian of the iron sheet and the energy harvesting voltage of the Halbach array is consistent with the simulation results. When the radian of the iron sheet was changed from 60 to 40°, the maximum energy harvesting voltage was basically unchanged. When the radian of the iron sheet continued to decrease, the maximum energy harvesting voltage also decreased. Through the transformation of the LTC-3108 circuit, the voltage can be stabilized to 2.8 V.

Figure 16 shows the time required to boost the voltage to 2.8 V under different radians of iron sheets. As can be seen from the figure, as the radian of iron sheet gets bigger, the time it takes to reach a stable voltage decreases. Therefore, iron sheet plays an important role in practical harvesting application.

## 5. Conclusions

Iron is an excellent ferromagnetic material. In this paper, simulations and experimental tests have proved that iron sheet indeed improves the energy harvesting effect of the Halbach array energy-harvesting structure. The energy-harvesting efficiency of the Halbach array can be significantly improved when iron sheet is added into the Halbach array at an appropriate angle and distance. When the angle between the iron sheet and the Halbach array is in the middle of the two radial magnetic poles, the Halbach array harvester with the iron sheet has the best energy-harvesting effect. The radian of the iron sheet determines the closure of the Halbach array. The larger the radian is, the higher the density magnetic lines are, and the better the energy-harvesting effect is. From completely enclosed to smaller and smaller radians, we found that the minimum radian of the iron sheet makes the least amount of sheet metal when the energy-harvesting effect is the greatest. The distance between the iron sheet and the Halbach array should be reduced as much as possible. With the decrease in distance, the energy-harvesting effect will increase. After adding the iron sheet the magnetic focusing effect of the Halbach array is enhanced, and the energy-harvesting efficiency is also improved and the structure volume is reduced. Because of its simple structure and easy miniaturization, it can be well applied to energy harvesting.

## Figures and Tables

**Figure 1 micromachines-12-01541-f001:**
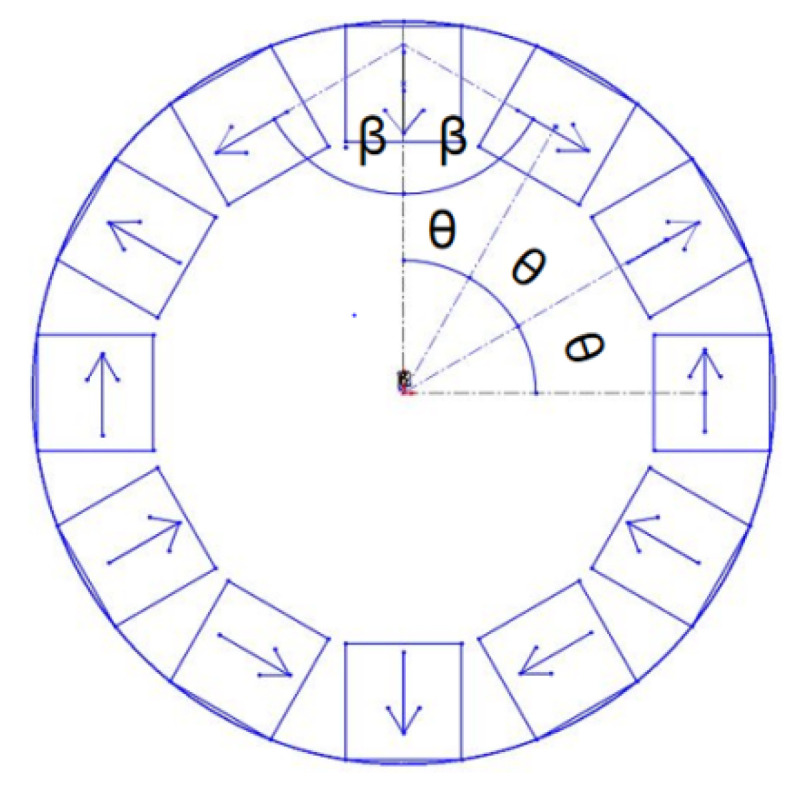
Permanent magnet arrangement of Halbach array.

**Figure 2 micromachines-12-01541-f002:**
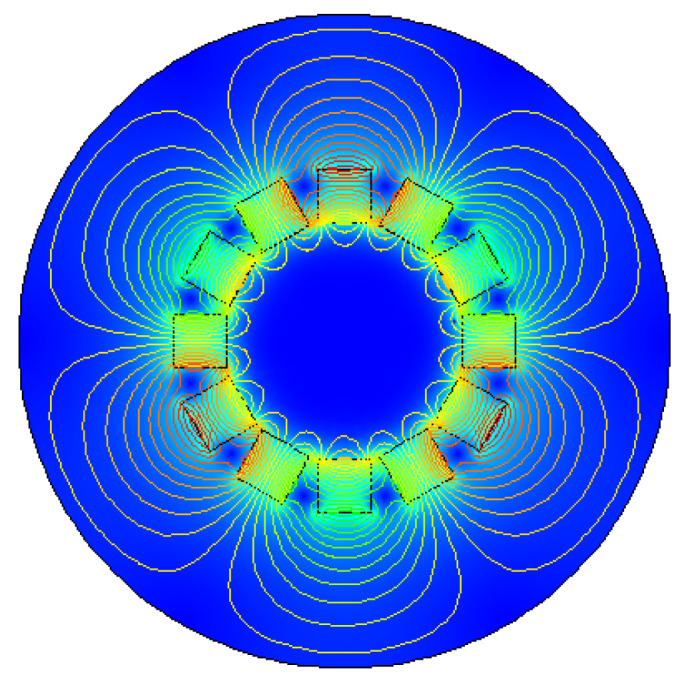
Halbach array magnetic field distribution diagram.

**Figure 3 micromachines-12-01541-f003:**
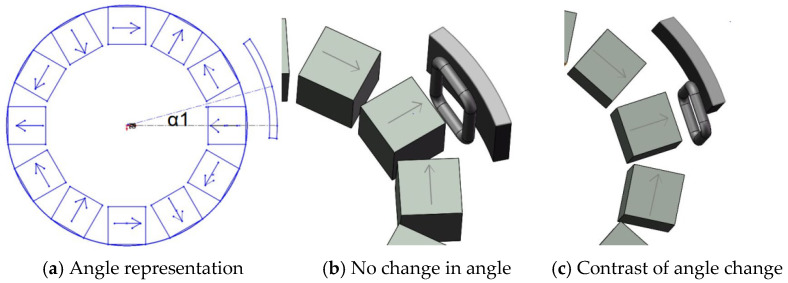
The angle between Halbach array and iron sheet.

**Figure 4 micromachines-12-01541-f004:**
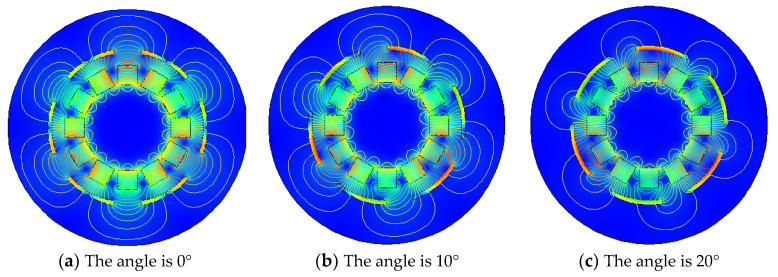
The magnetic field distribution diagram of iron sheet and Halbach array at different angles.

**Figure 5 micromachines-12-01541-f005:**
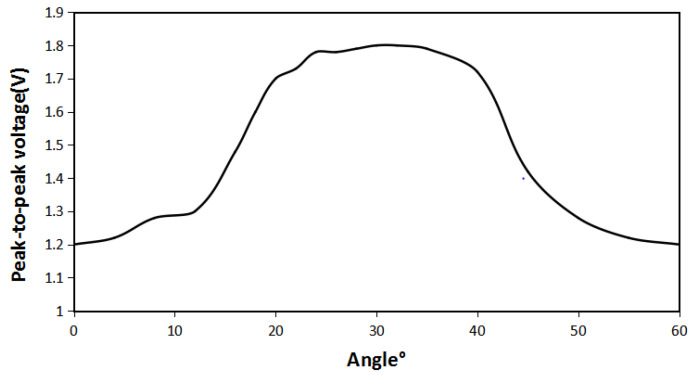
Peak-to-peak voltage of iron sheet and Halbach array rotor at different angles.

**Figure 6 micromachines-12-01541-f006:**
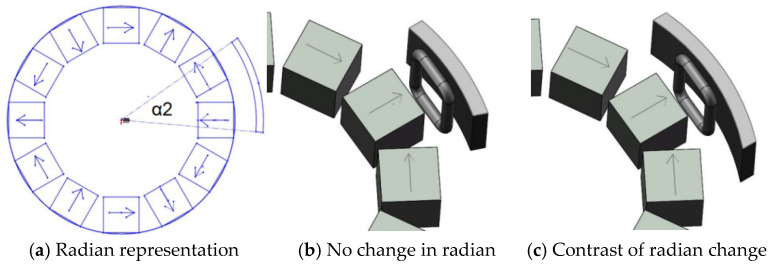
The radian of iron sheet.

**Figure 7 micromachines-12-01541-f007:**
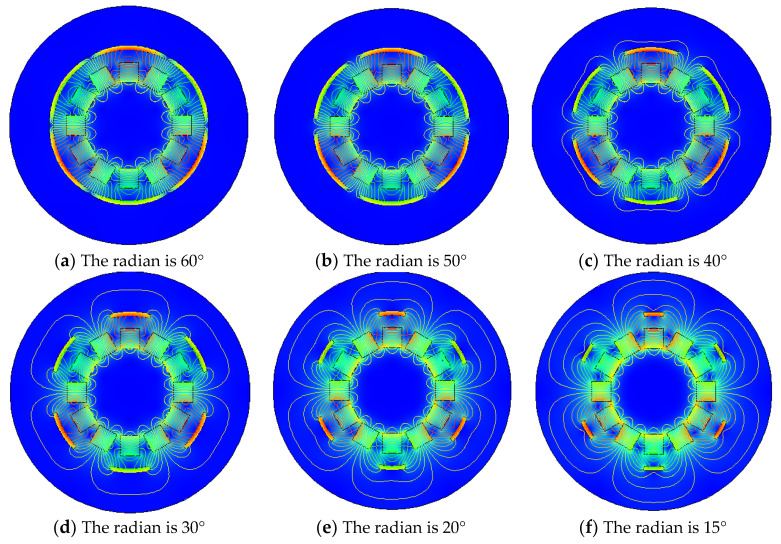
Magnetic field distribution of Halbach array under different iron sheet radians.

**Figure 8 micromachines-12-01541-f008:**
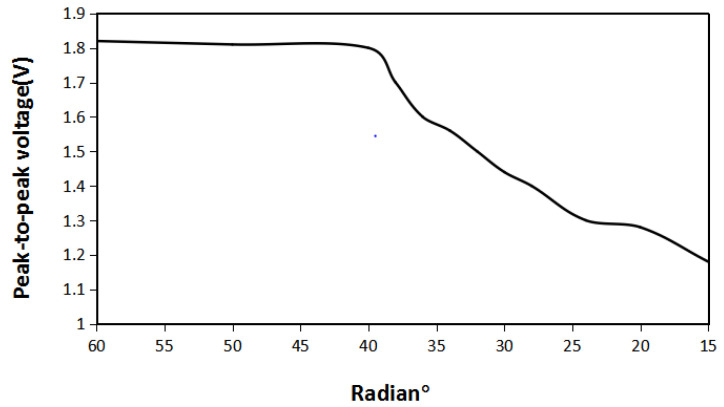
The maximum energy harvesting voltage of the Halbach array under different iron sheet radians.

**Figure 9 micromachines-12-01541-f009:**
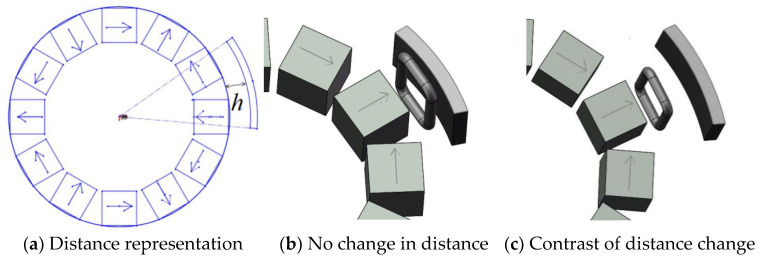
The distance between iron sheet and the Halbach array.

**Figure 10 micromachines-12-01541-f010:**
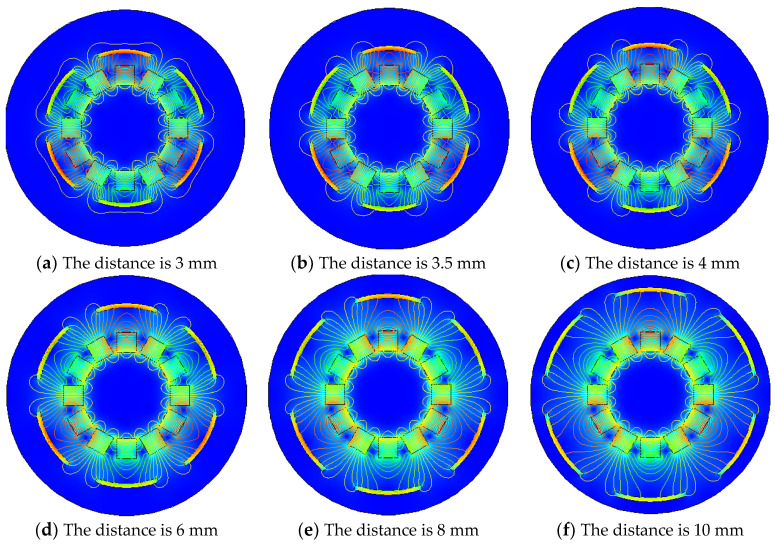
Magnetic field distribution of iron sheet and Halbach array at different distances.

**Figure 11 micromachines-12-01541-f011:**
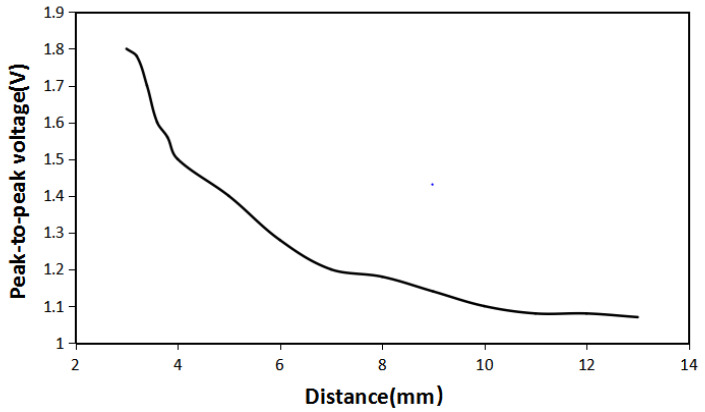
The maximum energy harvesting voltage of iron sheet and the Halbach array at different distance.

**Figure 12 micromachines-12-01541-f012:**
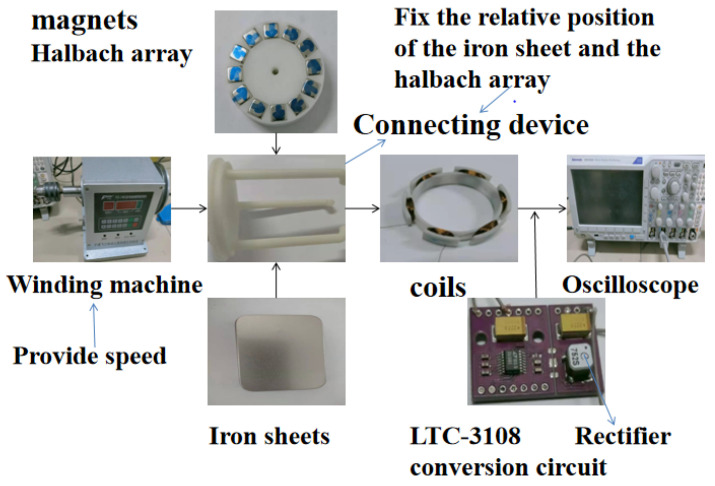
Devices used in the experiment.

**Figure 13 micromachines-12-01541-f013:**
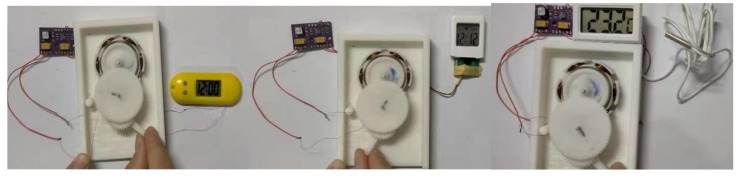
Energy supply experiment of Halbach array to different devices.

**Figure 14 micromachines-12-01541-f014:**
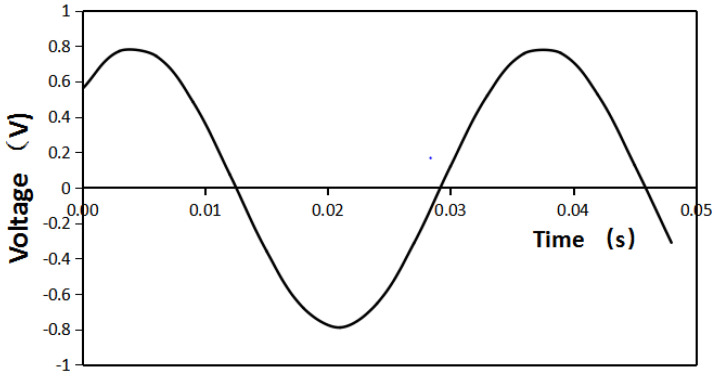
Voltage measured without iron sheet.

**Figure 15 micromachines-12-01541-f015:**
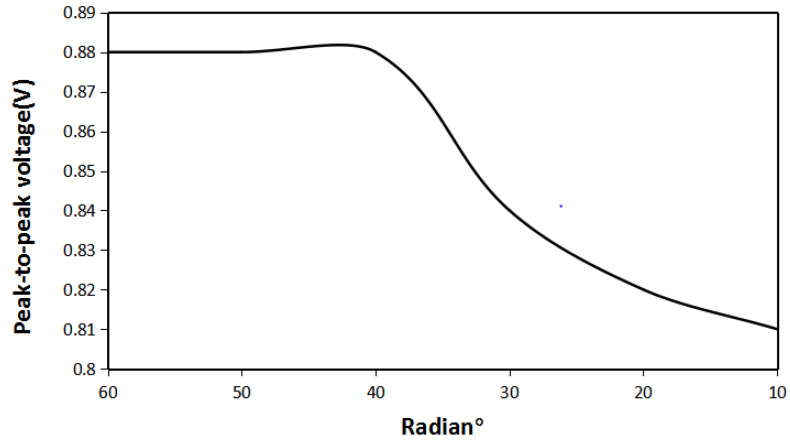
The relationship between the radian of iron sheet and the maximum voltage.

**Figure 16 micromachines-12-01541-f016:**
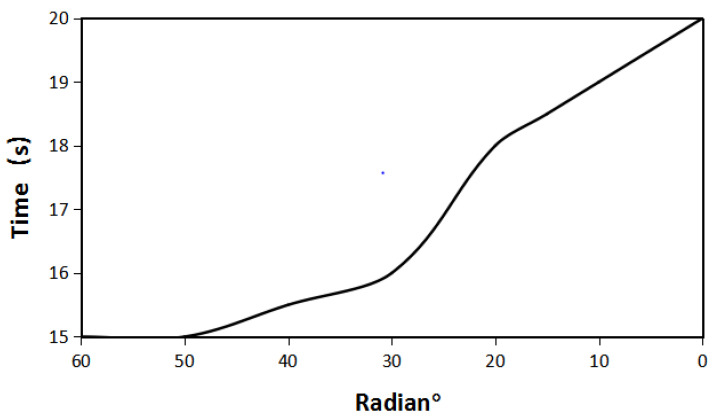
The time required to boost the voltage to 2.8 V under different iron radians.

**Table 1 micromachines-12-01541-t001:** Parameters used in the experiment.

Rectangular permanent magnet	5 mm × 5 mm × 5 mm
The radius of the Halbach array rotor	16 mm
The thickness of the iron sheet	1 mm
Permanent magnet residual magnetic flux density	0.84 T

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
