# Peer review of "Analysis of the Influence of Ferromagnetic Material on the Output Characteristics of Halbach Array Energy-Harvesting Structure"

_micromachines, 2021, doi:10.3390/mi12121541_

Round 1

Reviewer 1 Report

The authors study an iron sheet for Halbach array for energy harvesting. While energy harvesting is an important research topic, it is not clear to me what is new in this particular paper. As stated by the authors, the Halbach array has been known since 1979, and energy harvesting structures have been discussed. Therefore, I recommend that the authors significantly revise the manuscript, with a clear discussion of everything in the literature and what new aspects this particular paper brings to the field.

On the non-scientific side, I did not understand what are they harvesting. They mention a harvesting efficiency but never quoted an efficiency value. They have also noted a 1.8x jump in voltage that becomes 1.8 volts, but it is unclear what this jump represents.

On the scientific side, I did not understand the role of iron. What materials parameters play a role in the device's performance? What happens if I use a different material? Is iron is the best one can use? Why is there a big difference between theory and experiment?

The authors must cite everything that has discussed Halbach array for energy harvesting applications, e.g., MDPI Energies 2021, 14, 6094, and discuss how this paper is different.

Also, discuss the emerging magnetic array-based energy harvesting technologies that work efficiently in nanoscales. e.g., Nature Communications 12, 2924 (2021) and Appl. Phys. Lett. 118, 052408 (2021).

Reviewer 2 Report

This paper verifies the influence of ferromagnetic materials on the output of the Halbach array energy harvesting by exploring the angle between iron sheet and Halbach array, radian size of iron sheet and distance between iron sheet and Halbach array. By optimizing the parameters of the iron sheet can change the magnetic field distribution of the Halbach array and increase energy harvesting efficiency significantly. The manuscript is not ready for publication and requires a minor revision. The specific comments are listed as below:

  1. In the effect of iron sheet on Halbach array part, there is only the magnetic field distribution of the Halbach array without iron sheet in figure 2. Why it can be seen from figure 2 that the sinusoidal distribution of the synthetic magnetic field is better than that of none iron sheet?
  2. In the manuscript, only in the Experimental verification part there are some description of the energy harvesting structure, you can add a structure diagram of the energy harvester to make readers can better understand the working principle of the device.
  3. The Experimental verification part should add the experiment of different angles and distance between the Halbach array and the iron sheet to verify the simulation results.

Author Response

  1. In Figure 2, it is the magnetic field distribution diagram of the halbach array when there is no iron sheet, which has been corrected in the paper。
  2. In the revised paper, the working principle diagram of the device is added
  3. In the experiment, the device used to rotate the halbach array and the iron sheet is a winding machine,In the actual measurement, due to the accuracy of the equipment, a large gap between the iron sheet and the halbach needs to be reserved to ensure that it will not affect the coil.Therefore, the measurement of distance and angle has not been carried out, and improvements will be made in this area in the following.